# Simplified Synthesis and Stability Assessment of Aflatoxin B_1_-Lysine and Aflatoxin G_1_-Lysine

**DOI:** 10.3390/toxins14010056

**Published:** 2022-01-14

**Authors:** Justin B. Renaud, Jacob P. Walsh, Mark W. Sumarah

**Affiliations:** 1London Research and Development Centre, Agriculture and Agri-Food Canada, London, ON N5V 4T3, Canada; jwalsh69@uwo.ca; 2Department of Chemistry, University of Western Ontario, London, ON N6A 3K7, Canada

**Keywords:** aflatoxin, biomarker, aflatoxin B_1_-lysine, aflatoxin G_1_-lysine

## Abstract

Aflatoxins B_1_ (AFB_1_) and G_1_ (AFG_1_) are carcinogenic mycotoxins that contaminate crops such as maize and groundnuts worldwide. The broadly accepted method to assess chronic human aflatoxin exposure is by quantifying the amount of aflatoxin adducted to human serum albumin. This has been reported using ELISA, HPLC, or LC-MS/MS to measure the amount of AFB_1_-lysine released after proteolysis of serum albumin. LC-MS/MS is the most accurate method but requires both isotopically labelled and unlabelled AFB_1_-lysine standards, which are not commercially available. In this work, we report a simplified synthetic route to produce unlabelled, deuterated and ^13^C_6_ ^15^N_2_ labelled aflatoxin B_1_-lysine and for the first-time aflatoxin G_1_-lysine. Additionally, we report on the stability of these compounds during storage. This simplified synthetic approach will make the production of these important standards more feasible for laboratories performing aflatoxin exposure studies.

## 1. Introduction

Aflatoxins are the most important mycotoxins from a human health perspective, especially in developing countries. It is estimated that over 500 million people in parts of Asia, Latin-American, and sub-Saharan Africa are being exposed to aflatoxin B_1_ (AFB_1_) mainly through the consumption of maize and groundnuts infected with *Aspergillus flavus* and/or *A. parasiticus* [1,2,3].

AFB_1_ is highly mutagenic, arising from epoxidation of the 8,9 vinyl double bond by human liver P450 enzymes (CYP3A4, CYP1A2, and CYP3A5 in some individuals) [4]. AFB_1_-8,9 exo-epoxide is considered to be one of the most carcinogenic compounds known [5]. It efficiently chelates between DNA base pairs to react with guanine residues leading to base pair mutations [6]. Ingested AFB_1_ can also be transformed enzymatically into hydroxylated metabolites, such as AFM_1_, which is far less mutagenic, although it is genotoxic [7,8] or to AFB_1_-N^7^-guanine, which is excreted in urine. The carcinogen AFB_1_-8,9 *exo* epoxide hydrolyzes rapidly under aqueous conditions to AFB_1_-dihydrodiol which is in equilibrium with AFB_1_-dialdehyde [9] (Figure 1). AFB_1_-dialdehyde is highly reactive towards amines [10] and forms covalent adducts with lysine residues within human serum albumin [11,12,13].

Aflatoxin exposure can be determined by monitoring the urinary metabolites AFM_1_ and AFB_1_–N^7^-guanine; however, this only accounts for exposures for the preceding day or two. Although these urinary metabolites have frequently been utilized as biomarkers for hepatocellular carcinoma risk, these associations are weak [14,15]. The most robust biomarker for aflatoxin exposure is the AFB_1_-albumin adduct (AFB_1_-alb) [16]; the half-life of serum albumin (~21 d) [17] makes it an average of chronic exposure during the past month [1]. Although unmetabolized AFB_1_ can be detected in the urine, it is not believed to be strongly correlated with aflatoxin exposure [15].

Given the importance of aflatoxin on a global health scale, accurate and standardized methods to quantify the AFB_1_-adduct are critical, yet there are key shortcomings that still need to be addressed. Specifically, the lack of commercially available analytical aflatoxin biomarker standards and well characterized quality control reference material are major obstacles when comparing and interpreting findings across multiple studies.

The AFB_1_-alb adduct has been measured using ELISA [18,19], HPLC-fluorescence [20,21], and LC-MS/MS [22] by quantifying AFB_1_-Lys released by enzymatic digestion. ELISA based measurements have historically reported higher concentrations than HPLC-based approaches or LC-MS/MS by a factor of 2.6 [23]. Although the reason for this discrepancy between methods is unknown, cross-reactivity with other aflatoxin-serum adducts may be responsible. Similar to AFB_1_, aflatoxin G_1_ produced by *A. parasiticus* is an order of magnitude less carcinogenic in animals than B1 (Ayers et al. 1971; Butler et al. 1969). Aflatoxin G1 can also be metabolized to a mutagenic epoxide [24] and can result in formation of AFG_1_-alb adducts, although they have yet to be reported in humans [25].

AFB_2a_ is produced via acid hydrolysis of AFB_1_ (Figure 1) [26,27,28], and has been detected as a major urinary metabolite in some human studies [29]. Since it is missing the critical 8,9 vinyl double bond, AFB_2a_ is significantly less carcinogenic than AFB_1_ [7]. However, similar to the AFB_1_-epoxide, it can also rearrange to an amine reactive dialdehyde structure [30,31]. As with AFG_1_, albumin adducts of AFB_2a_ have not yet been detected in human populations, although it has been suggested they exist [32].

Direct analytical methods such as LC-MS/MS and HPLC require an analytical standard, which to date has been exclusively AFB_1_-Lys, and therefore, there are no data regarding the presence of other albumin adducts arising from AFB_2a_, AFG_1_, or even AFG_2a_. The synthesis of AFB_1_-Lys (reviewed in [33]), has generally followed a synthetic approach that mimics the in vivo route (Figure 1). First, AFB_1_ is epoxidized with either dimethyldioxirane [24,34], or *m*-chloroperbenzoic acid [35,36] to form the carcinogen AFB_1_-epoxide. Synthesis of this intermediate is not without risk and requires careful handling. Following epoxidation, AFB_1_-epoxide is then hydrolyzed to AFB_1_-dialdehyde and coupled to free lysine [36]. In previous synthetic routes, the Nα group of lysine has been protected by acetylation [10,37] or complexation with copper [34,38]. Once purified, the concentration of the analytical standard is determined by spectrophotometry using a molar attenuation coefficient *ε*_399_ 25,400/M·cm or *ε*_400_ 30,866/M·cm at pH 7.0 and 7.4, respectively [11,37]. To our knowledge, these molar attenuation coefficients are the sole basis for determining synthesized standard concentrations in subsequent studies, and it is unknown how any oxidized or degraded AFB_1_-Lys present could affect these values.

Recently, a study examining potential interactions between AFB_2a_ and a number of free amino acids showed that AFB_2a_ could be coupled with lysine to form AFB_2a_-Lys [31]. The authors showed that the addition of a strong oxidizing agent could convert AFB_2a_-Lys to AFB_1_-Lys. Although this was performed at a low concentration, this finding is significant for synthetic efforts since it would enable the highly carcinogenic AFB_1_-epoxide intermediate to be bypassed.

In this work, we address one of the main shortcomings facing the determination of human exposure to AFB_1_. We undertook a systematic approach to optimize a safer and simplified synthetic route for the production of AFB_1_-Lys. We also expanded the synthesis from AFB_1_-Lys to a total of four analytes that should be monitored in a single biological sample: AFB_1_-Lys, AFB_2a_-lysine, AFG_1_-Lys, and AFG_2a_-Lys. In addition, we generated isotopically labelled materials with Lysine-D_4_ and Lysine ^13^C_6_, ^15^N_2_ to serve as internal standards. We then performed a stability assay to determine the best storage conditions for these important standards.

## 2. Results

### 2.1. Conversion of AFB_1_ to AFB_2a_ and AFG_1_ to AFG_2a_

The efficient conversion of AFB_1_ to the hemiacetal AFB_2a_ using 0.1 M citric acid at 28 °C in the range 24–48 h was reported by Ciegler et al. [28]. The authors also showed that hydrolysis was likely occurring with AFG_1_, but they could not isolate a sufficient quantity for confirmation. The hemiacetal AFB_2a_ can react readily with the amino groups of amino acids and proteins [39]. AFB_2a_ undergoes a structural rearrangement at alkaline pH (pKa 7.1) to form a phenolate ion with two aldehyde groups. These aldehyde groups are necessary for AFB_2a_ to couple with free amino groups [30].

We employed HCl as the acid source because its low vapor pressure allowed for effective removal by evaporation during the synthesis of AFB_2a_. Over 48 h at 45 °C, AFB_1_ was readily converted to AFB_2a_ under mild acidic conditions; the conversion of both AFB_1_ and AFG_1_ to AFB_2a_ and AFG_2a_, were >95% (Figure 1)

### 2.2. Coupling of AFB_2a_ or AFG_2a_ to Lysine

The equivalence point of the hemiacetal AFB_2a_ and AFB_2a_-dialdehdye (Figure 1) was measured and shown to occur at pH 7.1 [30]. As the dialdehyde form is necessary for the coupling of AFB_2a_ to amines, reaction conditions within the pH range 7–11 were investigated.

Following the addition of lysine to AFB_2a_ at a 2:1 molar ratio, the aqueous solutions developed into an intense yellow colour. Rushing et al. (2017) identified this coupling to be first order with respect to both reactants and that the reaction rate increased with increasing pH [31]. Lysine contains two primary amines that are able to couple with AFB_2a_ dialdehyde. In order to mimic the reactivity of AFB_1_ dialdehyde within serum albumin lysine residues, it is necessary to direct the reaction towards the ε-amino group (Figure 2). This was previously accomplished with AFB_1_-dialdehyde by protecting the α-amino group by the use of Nα-acetyl-lysine [10,37] or complexation of lysine with copper [34,38]. Reaction with Nαacetyl-lysine necessitates an additional treatment with acylase I. More recently, Sass and et al. showed that when AFB_1_-dialdehyde was reacted with free lysine, the major product was the desired ε-group [36]. Here we closely monitored the effects of pH on the formation of either the α or ε-coupled AFB_2a_-Lys.

At pH 8.0, we observed the appearance of two isobaric peaks corresponding to the predicted formula of AFB_2a_-Lys, C_23_H_24_N_2_O_7_ (Figure 2). The minor and earlier eluting peak corresponded to the AFB_2a_-dialdehyde coupled to the α-amino group, whereas the major peak is the desired ε-linked product. The order of elution was determined by Rushing et al. (2017) who showed the later eluting peak was exclusively formed with acetyl-lysine reactant, whereas two peaks are observed with free lysine [31].

Figure 2c shows the ratio of the α/*ε* coupled AFB_2a_-lysines as a function of pH. At pH 7–8.5, the ratio is fairly constant at 13 ± 0.2%. This ratio decreases to 7 ± 0.5% at pH ≥ 9.5. Although not dramatic, this trend closely follows the ratio of protonation for α-NH_3_^+^/ε-NH_3_^+^ (Figure 2c). At pH ≤ 9.5, both amino groups of lysine are nearly fully protonated, yet the ε-NH_3_^+^ is still highly favoured meaning that steric factors may greatly influence this preference of reactivity (Figure 2).

Using a pH of 10.0 in subsequent experiments, a series of molar ratios between AFB_2a_ and lysine were investigated. In the absence of lysine, the λ_max_ of AFB_2a_ is 399 nm. Molar ratios of 1:1, 1:2, 1:4, and 1:10 AFB_2a_:lysine were mixed and the UV–VIS signature taken after 5 min. A clear progression of λ_max_ was observed, with the 1:10 mixture having a λ_max_ of 406 nm (Figure 3a). Following incubation at room temperature for 30 min, all the AFB_2a_:lysine ratios shifted to λ_max_ values between 406 and 408 nm (Figure 3b). When these solutions were tested by LC-MS, unreacted AFB_2a_ was not detected in any of the samples, while the intensity of the AFB_2a_-Lys product was slightly lower in the 1:1 mixture compared to mixtures with excess lysine (Appendix A). Therefore, for proceeding reactions, a 1:2 ratio of AFB_2a_:lysine was used.

### 2.3. Oxidation of AFB_2a_-Lys to AFB_1_-Lys

Rushing et al. (2017) used Oxone^®^ to demonstrate that AFB_2a_-Lys could be converted to AFB_1_-Lys via the addition of single oxygen atom [31]. It has been suggested that if AFB_2a_-alb adducts are present in vivo, its oxidation via natural biochemical processes may result in the formation of AFB_1_-alb [32], bypassing the formation of AFB_1_-epoxide (Figure 1). Therefore, in addition to optimizing this final oxidation step using Oxone^®^, we also investigated H_2_O_2_ due to its presence in cells as a short-lived metabolic by-product.

A small amount of AFB_2a_-Lys (10 µg AFB_1_ equivalents) was dried down and reconstituted in 195 µL of the 9 various solvents and buffers. A molar ratio of 1:10 AFB_2a_-Lys:H_2_O_2_ resulted in limited observable changes whereas 1:100 AFB_2a_-Lys:H_2_O_2_ and 1:1 AFB_2a_-Lys:Oxone^®^ resulted in the appearance of new products. In addition to unreacted AFB_2a_-Lys, and the desired AFB_1_-Lys (I), AFB_1_-Lys with an additional oxygen (Figure 3, II) and AFB_1_-Lys with an additional double bond and oxygen (Figure 3, III) were significant by-products. These unknown oxidation products were examined by high resolution LC-MS/MS (Figure 4) and their structures are proposed in Figure 3.

To systematically determine the best oxidizing agent for this reaction, we examined by LC-MS, both Oxone^®^ and H_2_O_2_ under various reaction conditions (Figure 5 and Figure 6).

For Oxone^®^, a significant level of oxidation occurs at a molar ratio of 1:1, however this reaction is pH dependant. At pH 8, a maximum amount of AFB_1_-Lys is produced, while by-products II and III still account for approximately 30% of the products generated (Figure 5a). Increasing the amount of Oxone^®^ used results in a decrease in unreacted AFB_2a_-Lys, however there is also an increase in the by-products with an added oxygen atom. Additionally, although the proportion of the desired AFB_1_-Lys product does increase with increasing Oxone^®^, the overall signal decreases (Figure 5c). This may be the result of the formation of pyrrole products that are able to polymerize [40].

When H_2_O_2_ is added at a 1:10 ratio, limited conversion of AFB_2a_-Lys to AFB_1_-Lys was observed, however at a higher ratio, there was efficient oxidation. Notably, when performed in methanol, the reactant was nearly completely consumed and converted mostly to AFB_1_-Lys. The unwanted oxidized by-products (Figure 3), were also minimized under these conditions. Increasing the pH of the reaction solution, did coincide with an increased quantity of oxidized by-products (Figure 6b). Unlike with Oxone^®^, H_2_O_2_ added to a methanolic solution did not greatly decrease the overall signal of the desired product AFB_1_-Lys. This may be due to conditions which do not favour polymerization (Figure 6c). Therefore, we determined that the most efficient route of converting AFB_2a_-Lys to AFB_1_-lys is via the addition of H_2_O_2_ in a methanolic solution.

### 2.4. Synthesis of Isotopically Labelled Standards

The analysis of AFB_1_-Lys by LC-MS/MS relies on isotopically labelled internal standards to account for sample matrix effects and analyte loss during sample cleanup. Previous studies used lysine-D_4_ [38], whereas we used ^13^C_6_,^15^N_2_ lysine [34]. In this work, using the optimized synthetic route described above, we were able to efficiently produce both AFB_1_-Lys and AFG_1_-Lys using unlabelled, deuterated and ^13^C_6_,^15^N_2_ lysine for a total of six standards (Figure 7). A minimum of one internal standard per analyte is required to quantify these analytes in samples, however having two distinct isotopically labelled standards will allow for an additional performance step such as SPE or the Pronase^®^ digestion to be monitored.

### 2.5. Quantitative NMR

To our knowledge, the concentrations of synthetically derived AFB_1_-Lys were determined by the molar attenuation coefficients described by Sabbioni (*ε*_399_ 25,400/M·cm at pH 7.0 [11] and *ε*_400_ 30,866/M·cm pH 7.4 [37]). During our first attempts to dissolve the isolated standards at concentrations at or above 1 mg/mL, we observed the formation of precipitates with methanol, D_2_O, and D_2_O with 0.01 M K_2_DPO_4_. These precipitates could not be resolubilized by changing the solvent, or with heating and sonication. Therefore, we aimed to perform qNMR at lower concentrations (0.5–1 mg/mL) and using a mixed solvent system that contained 70% D_2_O and 30% MeCN-D_3_.

Although spectroscopic quantitation is more straightforward than qNMR, if different groups synthesize standards that contain oxidized impurities, these molar attenuation coefficients may not accurately describe the concentration of synthetic AFB_1_-Lys standards. Furthermore, should the standards undergo oxidation or degradation while in storage, checking the standard by spectrophotometry alone could also lead to inaccuracies. In this work, we undertook qNMR, using maleic acid as an internal standard. (Table 1).

We observed good agreement between concentrations of AFB_1_-Lys measured with UV–VIS and qNMR. This is significant as to our knowledge, all studies that have measured AFB_1_-Lys in samples have relied on standards that have been characterized by the published molar attenuation constant and therefore, are likely accurate. This contrasts with our results for AFG_1_-Lys, where we observed a significant disagreement between UV–VIS and qNMR derived concentrations. The UV–Vis AFG_1_-Lys concentration was lower (%) than the concentrations determined by qNMR; 1.89 vs. 0.634 mM, respectively. We re-examined this disagreement by recording the mass of the dried AFG_1_-Lys material in a pre-weighed vial, and resolubilizing the solution in 25% acetonitrile at 1 mg/mL. Similar to the qNMR comparison, UV–Vis concentrations were significantly less than what was expected based on the measured masses. The solution of AFG_1_-Lys that was 1 mg/mL by mass, gave a measured concentration of only 0.3 mg/mL by UV–Vis. Therefore, the molar attenuation constant previously reported for AFG_1_-Lys is likely not accurate and should not be used [25]. To our knowledge, AFG_1_-Lys has not been measured as a biomarker in any populations and therefore this discrepancy has likely had limited impact on any published studies.

### 2.6. Stability

The stability of AFB_1_-Lys was assessed in a 21 day assay under various conditions. The standard at 50 ng/mL was initially tested in various solvents, where it was found that it rapidly degraded in DMSO and had limited solubility in acetonitrile. Therefore, these two conditions were omitted from the full 21-day assay. With the exception of being stored as a dried residue with PBS salts, after 14 days (Figure 8a), the analyte was stable in all tested conditions when kept at −20 °C or 4 °C. Conversely, at 25 °C, degradation was observed in all solvents with the exception of methanol. The most significant degradation was observed when the analyte was stored in PBS, as a dried residue, or a dried residue with PBS salts. Therefore, these synthetic AFB_1_-Lys and AFG_1_-Lys standards if possible should not be shipped or transported as a dry residue.

A closer inspection on the stability of AFB_1_-Lys over 21 days in 50% acetonitrile or methanol (Figure 8b,c) indicates good stability can be maintained at colder temperatures across the tested time frame. At lower concentrations, the standards could be stored in methanol, however based on observations that insoluble precipitates could form in methanol at higher concentrations, storing the standards in a 50:50 water:acetonitrile solution is preferred for long term storage.

## 3. Conclusions

In this study we simplified the synthesis of AFB_1_-Lys and presented the synthesis of AFG_1_-Lys for the first time. We also produced Lysine-D_4_ and Lysine ^13^C_6_, ^15^N_2_ labelled versions of both and studied their stability under a number of conditions. Access to these standards that are not commercially available is critical for conducting human aflatoxin exposure assessments. The next steps moving beyond this study are to develop a serum reference material and conduct a validation study comparing the currently available AFB_1_-Lys standards. This will ensure better cooperation between laboratories conducting aflatoxin exposure assessments and will also ensure that data are comparable from one study to the next, which is critical for studies conducted on human populations.

## 4. Materials and Methods

### 4.1. Materials

AFB_1_ and AFG_1_ were obtained from Toronto Research Chemicals (Toronto, ON, Canada). Sodium bicarbonate, 50% hydrogen peroxide solution, Oxone^®^, L-Lysine, L-Lysine-4,4,5,5-D_4_, hydrochloride, L-Lysine-^13^C,^15^N_2_ hydrochloride, D_2_O, acetonitrile-D_3_, and maleic acid (qNMR standard, TraceCert^®^) were obtained from Millipore Sigma (Oakville, ON, Canada). LC-MS grade solvents H_2_O, methanol, and acetonitrile were purchased from Fisher Scientific (Ottawa, ON, Canada).

LC-MS and high-resolution LC-MS/MS, all reactions were monitored using a Thermo Vanquish Duo UHPLC system coupled to a Thermo Altis triple quadrupole mass spectrometer. The UHPLC used an Agilent Zorbax C-18 column (50 × 2.1 mm; 1.8 µm) held at 35 °C, flow rate of 300 µL/min, aqueous mobile phase (A) of H_2_O +0.1% formic Acid (Fisher Optima™ LC/MS Grade) and organic mobile phase (B) of acetonitrile + 0.1% formic acid (Fisher, Optima™ LC/MS Grade) was used throughout. The gradient began with 2% B for 1.6 min before increasing to 100% over 2.4 min. B was held at 100% for 1 min before returning to 2% in 0.5 min. The analytes were monitored using an HESI source, in positive ionization full MS mode, with a range of *m/z* 150–800, scan rate of 1000 mass units/s, and resolution of 0.7 FWHM.

High resolution MS/MS spectra of analytes were obtained using a Q-Exactive Orbitrap mass spectrometer coupled to an Agilent 1290 UHPLC system. The chromatographic conditions were the same as previously described. Mass spectrometer analysis in positive mode was performed using the following conditions: heated electrospray ionization (HESI): capillary temperature, 400 °C; sheath gas, 17 arbitrary units; auxiliary gas, 8 units; probe heater temperature, 450 °C; S-Lens rf level, 45%; and capillary voltage, 3.9 kV. Analytes were analyzed by data-dependent analysis that included a full MS scan at 35,000 resolution, automatic gain control of 5 × 10^5^, followed by 3 MS/MS scans at 17,500 resolution, and normalized collision energy of 30.

### 4.2. Conversion of AFB_1_/AFG_1_ to AFB_2a_/AFG_2a_

AFB_1_ and AFG_1_ were converted to their ‘2a’ equivalents following the method of Ciegler et al. [28] with some modifications. One mg of each aflatoxin standard was placed in an amber HPLC vial and dissolved in 0.4 mL of LC-MS grade acetonitrile. The solution was vortexed for 10 s and sonicated in a 25 °C water bath for 3 min to ensure the compound was completely dissolved. A 0.5 mL aliquot of LC-MS grade H_2_O was added to the solution followed by 100 µL of 2 M HCl for a final acid concentration of 0.2 M. The vials were then placed on a thermomixer held at 42 °C with shaking at 300 rpm for 48 h. Following the incubation procedure, a 10 µL aliquot was removed and diluted in 990 µL of 50% acetonitrile and analyzed by LC-MS to ensure conversion was greater than 95% based on relative peak areas.

The resulting solutions were dried under a gentle stream of nitrogen gas at ambient temperature without additional heat. A 400 µL aliquot of acetonitrile was added to the residue and dried again to ensure that all residual acid was evaporated as it can negatively affect the coupling step. The dried aflatoxin residue was reconstituted in 50% acetonitrile at 1 mg/mL. Ten µL of this solution (10 µg AFB_1_ equivalents) was transferred into the wells of a costar^®^ 250 µL 96 well plates (Product: 3596, Corning Inc. Corning, New York, NY, USA), and dried under a gentle stream of nitrogen without heat. The subsequent coupling step was performed immediately after the residue was dried.

### 4.3. Coupling of Lysine to AFB_2a_

L-Lysine was dissolved in LC-MS grade H_2_O at a stock concentration of 1.9 mg/mL Solutions of 0.05 M sodium bicarbonate were pH adjusted with either HCl or NaOH to 7.0, 7.5, 8, 8.5, 9, 9.5, 10, 10.5, and 11 (±0.1). Then 195 µL of H_2_O, methanol, acetonitrile, or the buffers at the various pHs were added individually to the wells of a 96 well plate that contained 10 µg of the dried AFB_2a_ residue. A 5 µL aliquot of the 1.9 mg/mL lysine solution (9.4 µg) was added for a molar ratio of 1:2 AFB_2a_:Lysine and aspirated to ensure that any dried residue was solubilized. The plate was incubated at room temperature for 30 min. Ten µL aliquots were removed and diluted with 90 µL of 50% acetonitrile before being analyzed by LC-MS.

An equivalent experiment was performed in 0.05 M sodium bicarbonate buffer (pH 10.0 ± 0.1) and the volume of lysine was altered to obtain a molar ratios of AFB_2a_: Lysine of: 1:0 (25 µL H_2_O); 1:1 (2.5 µL Lys, 22.5 µL H_2_O); 1:2 (5 µL Lys, 20 µL H_2_O); 1:4 (10 µL Lys, 15 µL H_2_O), and 1:10 (25 µL Lys). The UV–VIS spectra in the range 220–800 nm were collected at 5 min and 30 min post addition of lysine using a Thermo Multiskan™ GO microplate spectrophotometer.

A scale up reaction was performed using 100 µg of dried AFB_2a_ material in an amber glass vial, to which 195 µL of 0.05 M sodium bicarbonate (pH 10.0 ± 0.1) was added and 5 µL of lysine at 19 mg/mL in H_2_O. The solution was incubated at room temperature for 30 min and a 10 µL aliquot was removed, diluted in 990 µL of 50% acetonitrile and analyzed by LC-MS. A similar approach to the scale up described above was used for the production of AFB_2a_-Lys-D_4_, AFB_2a_-Lys-^13^C_6_^15^N_2_, AFG_2a_-Lys, AFG_2a_-Lys-D_4_, and AFG_2a_-Lys-^13^C_6_^15^N_2_.

### 4.4. Oxidation of AFB_2a_-Lysine to AFB_1_-Lysine

Twenty µL (~10 µg AFB_1_ equivalents) of AFB_2a_-Lys prepared from the scale-up reaction described above was transferred into a 96 well plate and dried under a gentle stream of nitrogen without heat. 0.05 M dipotassium hydrogen phosphate (K_2_HPO_4_) was adjusted to pH 5 and pH 6; while 0.05 M sodium bicarbonate buffer was adjusted to pH 7, 8, 9, 10, and 11; all buffers were pH adjusted with either NaOH or HCl. The dried AFB_2a_-Lys was reconstituted in 195 µL of methanol, H_2_O, or the seven buffers listed above. The residue could not be fully resolubilized in pure acetonitrile, which was not included in this experiment.

Oxone^®^ (MW 152.2 Da) was dissolved in H_2_O at a concentration of 9.75 mg/mL (Oxone 10X). A portion of this solution was diluted with H_2_O to a concentration of 0.98 mg/mL (Oxone 1X). H_2_O_2_ (50% *v*/*v*, ρ = 1.197 g/mL) was diluted into H_2_O to a concentration of 18.3% (*v*/*v*) (H_2_O_2_ 1000x). This solution was further diluted to 1.8% (H_2_O_2_ 1X). Individually, 5 µL of either Oxone 10X, Oxone 1X, H_2_O_2_ 1000X, and H_2_O_2_ 100X were added to the dissolved AFB_2a_-Lys in the various solutions. The 1X represents a 1:1 molar ratio between the AFB_2a_-Lys and the Oxone^®^, the 10X solutions represent a 1:10 molar ratio. Similarly, 100x and 1000x represent a 1:100 and 1:1000 molar ratio between AFB_2a_-Lys and H_2_O_2_. Five µL of the stock 50% H_2_O_2_ solution added directly to the dried AFB_2a_-Lys material represented a 1:27,512 molar ratio between the AFB_2a_-Lys and the oxidant, respectively. The solutions with the oxidants were incubated at room temperature for 1 h with shaking at 300 rpm.

In a scale up reaction, dried AFB_2a_-Lys residue (100 µg AFB_1_ equiv.) in an amber glass vial, were resolubilized in 195 µL of MeOH and vortexed for 15 s. Five µL of 1000X H_2_O_2_ was added and the solution was incubated at room temperature for 1 h. The reaction mixture was then dried down under nitrogen gas without heat to remove any residual H_2_O_2_.

A similar approach to the scale up described above was used for the production of AFB_1_-lysine-D_4_, AFB_1_-Lys-^13^C_6_^15^N_2_, AFG_1_-Lys, AFG_1_-Lys-D_4_, AFG_1_-Lys-^13^C_6_^15^N_2_. A stepwise synthesis protocol is included in Appendix A.

### 4.5. Purification by HPLC

The reaction products were isolated with an Agilent 1200 HPLC system (Appendix A). Chromatographic conditions included an Eclipse XDB-C18 (9.4 × 250 mm, 5 μm; Agilent Technologies) column maintained at 35 °C with 100 µL injection volume. A 20-min gradient program consisting of a mobile phase of water with 0.1% trifluoroacetic acid (Sigma) (mobile phase A) and acetonitrile with 0.1% trifluoroacetic acid (mobile phase B) (Optima grade, Fisher Scientific) with a flow rate of 4 mL/min. The gradient began at 5% B, was held for 2 min, then increased to 45% B over 13 min, followed by an increase to 100% B over 0.5 min, then held at 100% B for 2.5 min. The method was completed by reconditioning the system to 5% B over 0.5 min and holding at 5% B for 1.5 min. Fractions were collected using time points established by analyzing the HPLC-UV spectra at 399 nm.

### 4.6. Quantitative NMR

Purified AFB_1_-Lys and AFG_1_-Lys were dried and reconstituted in D_2_O:acetonitrile-D_3_ (70:30 for AFB_1_-Lys and 75:25 for AFG_1_-Lys). A 25 µL aliquot was diluted in 975 µL of 0.1 M phosphate buffer (pH 7.4) and the concentrations were approximated by UV–VIS. The concentration of the AFB_1_-Lys solution approximated by UV–Vis using *ε*_400_ 30,866/M·cm [37] while AFG_1_-Lys was approximated using *ε*_413_ 27,783/M·cm [25]. Using this initial concentration, maleic acid (TraceCert^®^, Millipore, Sigma) dissolved in D_2_O was added to the aflatoxin standards to achieve a molar ratio with the aflatoxin-Lys adducts in the range 1:1–1:2 based on UV absorbance values.

Quantitative NMR (qNMR) data were obtained on a Varian 600 (I600) (Agilent Technologies, Santa Clara, CA, USA) with an Auto-XBD probe. The solvent system was D_2_O: acetonitrile-D_3_ (70:30 for AFB_1_-Lys and 75:25 for AFG_1_-Lys), and the NMR was locked to D_2_O. A five second relaxation delay was set, with a pulse angle of 45° with 256 scans at ambient temperature (22 °C). Quantitation was performed using five signals that were averaged and compared to the ^1^H NMR chemical shifts for maleic acid. The AFB_1_-Lys signals used were *δ* 2.57 (2H, integration: 2.00), *δ* 3.89 (2H, integration: 2.03), *δ* 3.37 (2H, integration: 1.97), *δ* 2.20 (1H, integration: 0.97), *δ* 3.45 (1H, integration: 1.04), averaged to 1.002 ± 0.024 per 1H, maleic acid was integrated for *δ* 3.20 (2H) or *δ* 1.60 (1H). For AFG_1_-Lys the signals used were *δ* 3.79 (1H, integration: 0.94), *δ* 3.45 (1H, integration: 1.00), *δ* 3.17 (2H, integration: 2.07), *δ* 1.78 (1H, integration: 1.06), *δ* 1.63 (2H, integration: 2.04), averaged to integration of 1.011 ± 0.041 per 1H, maleic acid was integrated for 0.670 (2H) or 0.335 (1H).

Using the qNMR derived concentrations, the solutions were diluted into 50% acetonitrile at a concentration of 10 µg/mL and transferred to Thermo Scientific Matrix barcoded storage tubes and stored at −80 °C.

### 4.7. Stability Assay

A solution of AFB_1_-Lys at 50 ng/mL was prepared in the following solvents: DMSO, 0.1M PBS (pH 7.4), methanol, H_2_O, and 50:50 acetonitrile: water. Aliquots of 50 µL at these concentrations were placed in individual polypropylene tubes and store at −20 °C, 4 °C, and 20 °C in darkness. Additionally, 50 µL solutions of 50 ng/mL AFB_1_-Lys in 0.1M PBS (pH 7.4) and H_2_O were placed into polypropylene tubes, and dried down using a centrivap and also placed at either −20 °C, 4 °C, or 20 °C.

At various time points, the tubes containing the AFB_1_-Lys were transferred to −80 °C and stored until the day of analysis. Prior to analysis, the samples were brought to room temperature and 10 µL of the 50 ng/mL solutions was transferred to fresh tubes and diluted with 90 µL of water. The peak areas of each time point at the various conditions and temperatures were normalized to the peak area of the *t*0 sample which was stored at −80 °C for the entirety of the assay.

## Data Availability

Not applicable.

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
