# Peer review of "Simplified Synthesis and Stability Assessment of Aflatoxin B_1_-Lysine and Aflatoxin G_1_-Lysine"

_toxins, 2022, doi:10.3390/toxins14010056_

Round 1

Reviewer 1 Report

Check the grammatical errors like line 23, exposed to instead of exposed the. 

Introduction section seems too lengthy and should be a bit brief. 

In methodology section, statistical analysis and quality control data is missing. 

Author Response

Check the grammatical errors like line 23, exposed to instead of
exposed the.
This grammatical error has been addressed, the manuscript has been reviewed and other grammatical errors have been addressed.

Introduction section seems too lengthy and should be a bit brief.
The authors feel that the introduction length was necessary to adequately address the previous work in this topic while also providing relevance to the presented study. For these reasons we believe the introduction should not be truncated, even if a bit lengthy.  Specifically, we believe that AFB2a-lysine may infact be present in real human exposure samples and this introduction could serve that future work.

In methodology section, statistical analysis and quality control data is missing. "

We believe that in this specific experimental procedure, there was not a need to compare and contrast the steps in a statistical manner. However, if the reviewer wishes to point out a specific section that could benefit from a statistical approach we would be happy to examine it.

Reviewer 2 Report

The study proposes a procedure to generate standards of aflatoxins adducted to lysine. The proposal is interesting and original. Work is well presented and methodology is appropriate. However, authors should review the following comments and suggestions:

-According to the information of section 4.4. “Oxidation of AFB2a-lysine to AFB1-lysine”, the molar ratio between the AFB2a-Lys and peroxide are incorrect.

In line 386 “Five µL of the stock 50% H2O2 solution added directly to the dried AFB2a-Lys material represented a 1:55 molar ratio between the AFB2a-Lys and the oxidant, respectively.” H2O2 (50% v/v, ρ = 1.197 g/mL), have 17.59M. In 5ul have 87.95 µmol. 10 µg AFB1 equivalents represent 0.032 µmol. So, the molar ratio should be 1:2748

It is the same for H2O2 10X, and H2O2 1X, were the ratio should be 1:1000 and 1:100 respectively.

-Chromatograms obtained in section 4.5 “Purification by HPLC”, should be added.

-In line 162, “When these solutions were tested by LC-MS, unreacted AFB2a was not detected in any of the samples, while the intensity of the AFB2a-Lys product was slightly lower in the 1:1 mixture compared to mixtures with excess lysine”. Chromatograms should be shown as supplemental material

-The Figure 6 b shows AFB2a-lys 1:65 H2O2 while the figure caption shows 1:55AFB2a-Lys: H2O2

-in line 273, correct limited solubility acetonitrile for limited solubility in acetonitrile

-in line 341, when you mentions “dried under a gentle stream of nitrogen gas without heat”, it mean at room temperature?

-in line 354, “carefully aspirated”, it mean that was pipetted up and down to mix a solution? (also named triturate).

-in line 367, after the dot, add a space.

Author Response

The study proposes a procedure to generate standards of aflatoxins adducted to lysine. The proposal is interesting and original. Work is well presented and methodology is appropriate. However, authors should review the following comments and suggestions:

-According to the information of section 4.4. “Oxidation of AFB2a-lysine to AFB1-lysine”, the molar ratio between the AFB2a-Lys and peroxide are incorrect.

In line 386 “Five µL of the stock 50% H2O2 solution added directly to the dried AFB2a-Lys material represented a 1:55 molar ratio between the AFB2a-Lys and the oxidant, respectively.” H2O2 (50% v/v, ρ = 1.197 g/mL), have 17.59M. In 5ul have 87.95 µmol. 10 µg AFB1 equivalents represent 0.032 µmol. So, the molar ratio should be 1:2748

It is the same for H2O2 10X, and H2O2 1X, were the ratio should be 1:1000 and 1:100 respectively.

We double checked the calculations and agree this was a mistake and thank the reviewer for catching this. 5uL of 50% H2O2 is a 1:2752 ratio and this has been changed. We have also changed the 10x, and 1x to 1000x and 100x respectively throughout the document and figures.

-Chromatograms obtained in section 4.5 “Purification by HPLC”, should be added.

Chromatogram has been added into supplementary section (Figue S2) and we have updated section 4.5.

-In line 162, “When these solutions were tested by LC-MS, unreacted AFB2a was not detected in any of the samples, while the intensity of the AFB2a-Lys product was slightly lower in the 1:1 mixture compared to mixtures with excess lysine”. Chromatograms should be shown as supplemental material

Chromatograms have been added to the supplementary section (Figure S1).

-The Figure 6 b shows AFB2a-lys 1:65 H2O2 while the figure caption shows 1:55AFB2a-Lys: H2O2

This has been corrected.

-in line 273, correct limited solubility acetonitrile for limited solubility in acetonitrile

The “in” has now been added.

-in line 341, when you mentions “dried under a gentle stream of nitrogen gas without heat”, it mean at room temperature?

“without heat” has been amended to “at ambient temperature without additional heat”.

-in line 354, “carefully aspirated”, it mean that was pipetted up and down to mix a solution? (also named triturate).

We have changed statement to increase clarity. It is now “…and carefully aspirated to ensure that any dried residue was solubilized”.

-in line 367, after the dot, add a space.

A space has been added after the dot.

Reviewer 3 Report

The manuscript presents the synthesis of AFB1-Lys, AFG1-Lys and related isotopically labeled standards. The data are interesting and can be useful during the development of LC-MS/MS quantitative protocols for AFB1-Lys and AFG1-Lys determination. The experiments were well-planned and the results are easy to follow. I have only minor comments.

Specific comments:

Line 312. The heading "LC-MS and high resolution LC-MS/MS" should be marked as a new subsection 4.2. Please re-number the subsections in 4. Materials and Methods.

Lines 313-315. Looking at the analytical column used and applied flow rate, it seems that UHPLC system was used instead of HPLC one. Please make sure on this point.

Lines 322 - 329.  Did the studies using the Orbitrap MS employ the same conditions of separation of analytes ( analytical column, gradient settings) as it was described for triple quadrupole MS?

Line 342. Please change "MeCN" to "acetonitrile".

Line 347. Please add information about the storage conditions of the obtained dried material.

Section 2.2. How it was confirmed that the first eluting peak is α-amino linked AFB2a-Lys and the second is the ε-linked form (Figure 2)?

Line 161. Please change "406" to "406 nm".

Lines 197-204. Please clarify which method was used to examine the changes in the production of different AF-Lys derived products after Oxone treatment.

Please refer to the data included in the supplementary file in the text.

Author Response

The manuscript presents the synthesis of AFB1-Lys, AFG1-Lys and related isotopically labeled standards. The data are interesting and can be useful during the development of LC-MS/MS quantitative protocols for AFB1-Lys and AFG1-Lys determination. The experiments were well-planned and the results are easy to follow. I have only minor comments.

Specific comments:

Line 312. The heading "LC-MS and high resolution LC-MS/MS" should be marked as a new subsection 4.2. Please re-number the subsections in 4. Materials and Methods.

This has been corrected.

Lines 313-315. Looking at the analytical column used and applied flow rate, it seems that UHPLC system was used instead of HPLC one. Please make sure on this point.

The Thermo Vanquish Duo and the Agilent 1290 are UHPLC systems, the text has been updated to reflect this.

Lines 322 - 329.  Did the studies using the Orbitrap MS employ the same conditions of separation of analytes ( analytical column, gradient settings) as it was described for triple quadrupole MS?

It was operated under the same conditions as those used for the Vanquish Duo system. We have added a sentence to clarify this for the reader.

Line 342. Please change "MeCN" to "acetonitrile".

This has been updated to “acetonitrile” and other instances of the use MeCN have been reviewed as well.

Line 347. Please add information about the storage conditions of the obtained dried material.

The dried residue was coupled immediately with the lysine, however for clarity we added:

“The subsequent coupling step was performed immediately after the residue was dried.”

Section 2.2. How it was confirmed that the first eluting peak is α-amino linked AFB2a-Lys and the second is the ε-linked form (Figure 2)?

Retention time order was identified with respect to the study cited in reference “36” Sass, D.C.; Jager, A.V.; Tonin, F.G.; Rosim, R.E.; Constantino, M.G.; Oliveira, C.A.F. Synthesis and purification of the aflatoxin b1-lysine adduct. Toxin Reviews 2015, 34, 53-59

Line 161. Please change "406" to "406 nm".

Appropriate units have now been added.

Lines 197-204. Please clarify which method was used to examine the changes in the production of different AF-Lys derived products after Oxone treatment.

Results were examined by LC-MS relative intensities. To address this the text “by LC-MS” has been added to line 196, which introduces the paragraph in question.

Round 2

Reviewer 2 Report

The current version is accepted to be published